# Consistency Regularization
# for Variational Auto-Encoders

**Samarth Sinha**
Vector Institute
University of Toronto

**Adji B. Dieng**
Google Brain
Princeton University

## Abstract

Variational auto-encoders (VAES) are a powerful approach to unsupervised learning. They enable scalable approximate posterior inference in latent-variable models using variational inference (VI). A VAE posits a variational family parameterized by a deep neural network—called an *encoder*—that takes data as input. This encoder is shared across all the observations, which amortizes the cost of inference. However the encoder of a VAE has the undesirable property that it maps a given observation and a semantics-preserving transformation of it to different latent representations. This "inconsistency" of the encoder lowers the quality of the learned representations, especially for downstream tasks, and also negatively affects generalization. In this paper, we propose a regularization method to enforce consistency in VAES. The idea is to minimize the Kullback-Leibler (KL) divergence between the variational distribution when conditioning on the observation and the variational distribution when conditioning on a random semantic-preserving transformation of this observation. This regularization is applicable to any VAE. In our experiments we apply it to four different VAE variants on several benchmark datasets and found it always improves the quality of the learned representations but also leads to better generalization. In particular, when applied to the nouveau variational auto-encoder (NVAE), our regularization method yields state-of-the-art performance on MNIST, CIFAR-10, and CELEBA. We also applied our method to 3D data and found it learns representations of superior quality as measured by accuracy on a downstream classification task. Finally, we show our method can even outperform the triplet loss, an advanced and popular contrastive learning-based method for representation learning. [1]

## 1   Introduction

Variational auto-encoders (VAES) have significantly impacted research on unsupervised learning. They have been used in several areas, including density estimation (Kingma & Welling, 2013; Rezende et al., 2014), image generation (Gregor et al., 2015), text generation (Bowman et al., 2015; Fang et al., 2019), music generation (Roberts et al., 2018), topic modeling (Miao et al., 2016; Dieng et al., 2019), and recommendation systems (Liang et al., 2018). VAES have also been used for different representation learning problems such as semi-supervised learning (Kingma et al., 2014), anomaly detection (An & Cho, 2015; Zimmerer et al., 2018), language modeling Bowman et al. (2015), active learning (Sinha et al., 2019), continual learning (Achille et al., 2018), and motion prediction of agents (Walker et al., 2016). This widespread application of VAE representations makes it critical that we focus on improving them.

VAES extend deterministic auto-encoders to probabilistic generative modeling. The encoder of a VAE parameterizes an approximate posterior distribution over latent variables of a generative model. The encoder is shared between all observations, which amortizes the cost of posterior inference. Once fitted, the encoder of a VAE can be used to obtain low-dimensional representations of data, (e.g. for

---

[1]Code for this work can be found at `https://github.com/sinhasam/CRVAE`

35th Conference on Neural Information Processing Systems (NeurIPS 2021).

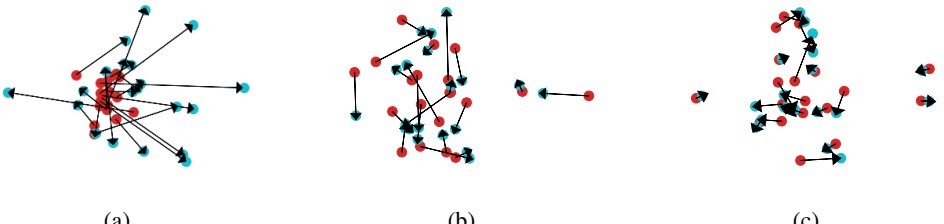

|  (a) | (b) | (c) |

**Figure 1:** Illustration of the *inconsistency* problem in VAES and how CR-VAES address this problem. The **red** dots correspond to the representations of few images from MNIST. The **blue** dots correspond to the representations of the transformed images. The transformations used here are rotations, translations, and scaling; they are semantics-preserving. The arrows connect the representations of any two pairs of an image and its transformation. The shorter the arrow, the better. **(a)**: The VAE maps the two sets of images to different areas in the latent space. **(b)**: Even when trained with the original dataset augmented with the transformed images, the VAE still maps the two sets of images to different parts in the latent space. **(c)**: The CR-VAE maps an image and its transformation to nearby areas in the latent space.

downstream tasks.) The quality of these representations is therefore very important to a successful application of VAES.

Researchers have looked at ways to improve the quality of the latent representations of VAES, often tackling the so-called *latent variable collapse* problem—in which the approximate posterior distribution induced by the encoder collapses to the prior over the latent variables (Bowman et al., 2015; Kim et al., 2018; Dieng et al., 2018; He et al., 2019; Fu et al., 2019).

In this paper, we focus on a different problem pertaining to the latent representations of VAES for image data. Indeed, the encoder of a fitted VAE tends to map an image and a semantics-preserving transformation of that image to different parts in the latent space. This "inconsistency" of the encoder affects the quality of the learned representations and generalization. We propose a method to enforce consistency in VAES. The idea is simple and consists in maximizing the likelihood of the images while minimizing the Kullback-Leibler (KL) divergence between the approximate posterior distribution induced by the encoder when conditioning on the image, on one hand, and its transformation, on the other hand. This regularization technique can be applied to any VAE variant to improve the quality of the learned representations and boost generalization performance. We call a VAE with this form of regularization, a consistency-regularized variational auto-encoder (CR-VAE).

Figure 1 illustrates the inconsistency problem of VAES and how CR-VAES address this problem on MNIST. The red dots are representations of a few images and the blue dots are the representations of their transformations. We applied semantics-preserving transformations: rotation, translation, and scaling. The VAE maps each image and its transformation to different parts in the latent space as evidenced by the long arrows connecting each pair (a). Even when we include the transformed images to the data and fit the VAE the inconsistency problem still occurs (b). The CR-VAE does not suffer from the inconsistency problem; it maps each image and its transformation to nearby areas in the latent space, as evidenced by the short arrows connecting each pair (c).

In our experiments (see Section 4), we apply the proposed technique to four VAE variants, the original VAE (Kingma & Welling, 2013), the importance-weighted auto-encoder (IWAE) (Burda et al., 2015), the $\beta$-VAE (Higgins et al., 2017), and the nouveau variational auto-encoder (NVAE) (Vahdat & Kautz, 2020). We found, on four different benchmark datasets, that CR-VAES always yield better representations and generalize better than their base VAES. In particular, consistency-regularized nouveau variational auto-encoders (CR-NVAES) yield state-of-the-art performance on MNIST and CIFAR-10. We also applied CR-VAES to 3D data where these conclusions still hold.

## 2    Method

We consider a latent-variable model $p_\theta(\mathbf{x}, \mathbf{z}) = p_\theta(\mathbf{x}|\mathbf{z}) \cdot p(\mathbf{z})$, where $\mathbf{x}$ denotes an observation and $\mathbf{z}$ is its associated latent variable. The marginal $p(\mathbf{z})$ is a prior over the latent variable and $p_\theta(\mathbf{x}|\mathbf{z})$

is an exponential family distribution whose natural parameter is a function of $\mathbf{z}$ parameterized by $\theta$, e.g. through a neural network. Our goal is to learn the parameters $\theta$ and a posterior distribution over the latent variables. The approach of VAEs is to maximize the evidence lower bound (ELBO), a lower bound on the log marginal likelihood of the data,

$$\mathcal{L}_{\text{VAE}} = \text{ELBO} = \mathbb{E}_{q_\phi(\mathbf{z}|\mathbf{x})} \left[ \log \left( \frac{p_\theta(\mathbf{x}, \mathbf{z})}{q_\phi(\mathbf{z}|\mathbf{x})} \right) \right] \tag{1}$$

where $q_\phi(\mathbf{z}|\mathbf{x})$ is an approximate posterior distribution over the latent variables. The idea of a VAE is to let the parameters of the distribution $q_\phi(\mathbf{z}|\mathbf{x})$ be given by the output of a neural network, with parameters $\phi$, that takes $\mathbf{x}$ as input. The parameters $\theta$ and $\phi$ are then jointly optimized by maximizing a Monte Carlo approximation of the ELBO using the reparameterization trick (Kingma & Welling, 2013).

Consider a semantics-preserving transformation $t(\tilde{\mathbf{x}}|\mathbf{x})$ of data $\mathbf{x}$ (e.g. rotation or translation for images.) A good representation learning algorithm should provide similar latent representations for $\mathbf{x}$ and $\tilde{\mathbf{x}}$. This is not the case for the VAE that maximizes Equation 1 and its variants. Once fit to data, the encoder of a VAE is unable to yield similar latent representations for a data $\mathbf{x}$ and its tranformation $\tilde{\mathbf{x}}$ (see Figure 1). This is because there is nothing in Equation 1 that forces this desideratum.

We now propose a regularization method that ensures *consistency* of the encoder of a VAE. We call a VAE with such a regularization a CR-VAE. The regularization proposed is applicable to many variants of the VAE such as the IWAE (Burda et al., 2015), the $\beta$-VAE (Higgins et al., 2017), and the NVAE (Vahdat & Kautz, 2020). In what follows, we use the standard VAE, the one that maximizes Equation 1, as the base VAE to regularize to illustrate the method.

Consider an image $\mathbf{x}$. Denote by $t(\tilde{\mathbf{x}}|\mathbf{x})$ the random process by which we generate $\tilde{\mathbf{x}}$, a semantics-preserving transformation of $\mathbf{x}$. We draw $\tilde{\mathbf{x}}$ from $t(\tilde{\mathbf{x}}|\mathbf{x})$ as follows:

$$\tilde{\mathbf{x}} \sim t(\tilde{\mathbf{x}}|\mathbf{x}) \iff \epsilon \sim p(\epsilon) \text{ and } \tilde{\mathbf{x}} = g(\mathbf{x}, \epsilon). \tag{2}$$

Here $g(\mathbf{x}, \epsilon)$ is a semantics-preserving transformation of the image $\mathbf{x}$, e.g. translation with random length $\epsilon$ drawn from $p(\epsilon) = \mathcal{U}[-\delta, \delta]$ for some threshold $\delta$. A CR-VAE then maximizes

$$\mathcal{L}_{\text{CR-VAE}}(\mathbf{x}) = \mathcal{L}_{\text{VAE}}(\mathbf{x}) + \mathbb{E}_{t(\tilde{\mathbf{x}}|\mathbf{x})} \left[ \mathcal{L}_{\text{VAE}}(\tilde{\mathbf{x}}) \right] - \lambda \cdot \mathcal{R}(\mathbf{x}, \phi) \tag{3}$$

where the regularization term $\mathcal{R}(\mathbf{x}, \phi)$ is

$$\mathcal{R}(\mathbf{x}, \phi) = \mathbb{E}_{t(\tilde{\mathbf{x}}|\mathbf{x})} \left[ \text{KL} \left( q_\phi(\mathbf{z}|\tilde{\mathbf{x}}) || q_\phi(\mathbf{z}|\mathbf{x}) \right) \right]. \tag{4}$$

Maximizing the objective in Equation 3 maximizes the likelihood of the data and their augmentations while enforcing consistency through $\mathcal{R}(\mathbf{x}, \phi)$. Minimizing $\mathcal{R}(\mathbf{x}, \phi)$, which only affects the encoder (with parameters $\phi$), forces each observation and the corresponding augmentations to lie close to each other in the latent space. The hyperparameter $\lambda \geq 0$ controls the strength of this constraint.

The objective in Equation 3 is intractable but we can easily approximate it using Monte Carlo with the reparameterization trick. In particular, we approximate the regularization term with one sample from $t(\tilde{\mathbf{x}}|\mathbf{x})$ and make the dependence to this sample explicit using the notation $\mathcal{R}(\mathbf{x}, \tilde{\mathbf{x}}, \phi)$. Algorithm 1 illustrates this in greater detail. Although we show the application of consistency regularization using the VAE that maximizes the ELBO, $\mathcal{L}_{\text{VAE}}(\cdot)$ in Equation 3 can be replaced with any VAE objective.

## 3 Related Work

Applying consistency regularization to VAEs, as we do in this paper, has not been previously explored. Consistency regularization is a widely used technique for semi-supervised learning (Bachman et al., 2014; Sajjadi et al., 2016; Laine & Aila, 2016; Miyato et al., 2018; Xie et al., 2019). The core idea behind consistency regularization for semi-supervised learning is to force classifiers to learn representations that are insensitive to semantics-preserving changes to images, so as to improve classification of unlabeled images. Examples of semantics-preserving changes used in the literature include rotation, zoom, translation, crop, or adversarial attacks. Consistency is often enforced by minimizing the $\mathbb{L}_2$ distance between a classifier's logit output for an image and the logit output for its semantics-preserving transformation (Sajjadi et al., 2016; Laine & Aila, 2016), or by minimizing

---

**Algorithm 1:** Consistency Regularization for Variational Autoencoders

---

**input** : Data $\mathbf{x}$, consistency regularization strength $\lambda$, latent space dimensionality K

Initialize parameters $\theta, \phi$

**for** iteration $t = 1, 2, \ldots$ **do**

    Draw minibatch of observations $\{\mathbf{x}_n\}_{n=1}^B$

    **for** $n = 1, \ldots, B$ **do**

        Transform the data: $\boldsymbol{\epsilon}_n \sim p(\boldsymbol{\epsilon}_n)$ and $\tilde{\mathbf{x}}_n = T(\mathbf{x}_n, \boldsymbol{\epsilon}_n)$

        Get variational mean and variance for the data:

            $\boldsymbol{\mu}_n = \mathbf{W}^\top \mathrm{NN}(\mathbf{x}_n; \phi) + \mathbf{a}$ and $\boldsymbol{\sigma}_n = \mathrm{softplus}(\mathbf{Q}^\top \mathrm{NN}(\mathbf{x}_n; \phi) + \mathbf{b})$

        Get $S$ samples from the variational distribution when conditioning on $\mathbf{x}_n$:

            $\boldsymbol{\eta}^{(s)} \sim \mathcal{N}(0, \mathbf{I})$ and $\mathbf{z}_n^{(s)} = \boldsymbol{\mu}_n + \eta^{(s)} \cdot \boldsymbol{\sigma}_n$ for $s = 1, \ldots, S$

        Get variational mean and variance for the transformed data:

            $\tilde{\boldsymbol{\mu}}_n = \mathbf{W}^\top \mathrm{NN}(\tilde{\mathbf{x}}_n; \phi) + \mathbf{a}$ and $\tilde{\boldsymbol{\sigma}}_n = \mathrm{softplus}(\mathbf{Q}^\top \mathrm{NN}(\tilde{\mathbf{x}}_n; \phi) + \mathbf{b})$

        Get $S$ samples from the variational distribution when conditioning on $\tilde{\mathbf{x}}_n$:

            $\boldsymbol{\eta}^{(s)} \sim \mathcal{N}(0, \mathbf{I})$ and $\tilde{\mathbf{z}}_n^{(s)} = \tilde{\boldsymbol{\mu}}_n + \eta^{(s)} \cdot \tilde{\boldsymbol{\sigma}}_n$ for $s = 1, \ldots, S$

    **end**

    Compute $\mathcal{L}_{\mathrm{VAE}}(\mathbf{x})$:

        $\mathcal{L}_{\mathrm{VAE}}(\mathbf{x}) \approx \frac{1}{B} \sum_{n=1}^B \frac{1}{S} \sum_{s=1}^S \left[ \log p_\theta(\mathbf{x}_n, \mathbf{z}_n^{(s)}) - \log q_\phi(\mathbf{z}_n^{(s)} | \mathbf{x}_n) \right]$

    Compute $\mathcal{L}_{\mathrm{VAE}}(\tilde{\mathbf{x}})$:

        $\mathcal{L}_{\mathrm{VAE}}(\tilde{\mathbf{x}}) \approx \frac{1}{B} \sum_{n=1}^B \frac{1}{S} \sum_{s=1}^S \left[ \log p_\theta(\tilde{\mathbf{x}}_n, \tilde{\mathbf{z}}_n^{(s)}) - \log q_\phi(\tilde{\mathbf{z}}_n^{(s)} | \tilde{\mathbf{x}}_n) \right]$

    Compute KL consistency regularizer:

        $\mathcal{R}(\mathbf{x}, \tilde{\mathbf{x}}, \phi) = \frac{1}{2} \sum_{k=1}^K \left( \frac{\tilde{\boldsymbol{\sigma}}_{nk}^2 + (\tilde{\boldsymbol{\mu}}_{nk} - \boldsymbol{\mu}_{nk})^2}{\boldsymbol{\sigma}_{nk}^2} - 1 + 2 \cdot \log \frac{\boldsymbol{\sigma}_{nk}}{\tilde{\boldsymbol{\sigma}}_{nk}} \right)$

    Compute final loss:

        $\mathcal{L}_{\mathrm{CR\text{-}VAE}}(\mathbf{x}) = \mathcal{L}_{\mathrm{VAE}}(\mathbf{x}) + \mathcal{L}_{\mathrm{VAE}}(\tilde{\mathbf{x}}) - \lambda \cdot \mathcal{R}(\mathbf{x}, \tilde{\mathbf{x}}, \phi)$

    Backpropagate through $\mathcal{L}(\mathbf{x}, \theta, \phi) = -\mathcal{L}_{\mathrm{CR\text{-}VAE}}(\mathbf{x})$ and take a gradient step for $\theta$ and $\phi$

**end**

---

the KL divergence between the classifier's label distribution induced by the image and that of its tranformation (Miyato et al., 2018; Xie et al., 2019).

More recently, consistency regularization has been applied to generative adversarial networks (GANS) (Goodfellow et al., 2014). Indeed Wei et al. (2018) and Zhang et al. (2020) show that applying consistency regularization on the discriminator of a GAN—also a classifier—can substantially improve its performance.

The idea we develop in this paper differs from the works above in two ways. First, it applies consistency regularization to VAES for image data. Second, it leverages consistency regularization, not in the label or logit space, as done in the works mentioned above, but in the latent space.

Although different, consistency regularization for VAES relates to works that study ways to constrain the sensitivity of encoders to various perturbations. For example, denoising auto-encoders (DAES) and their variants (Vincent et al., 2008, 2010) corrupt an image $\mathbf{x}$ into $\mathbf{x}'$, typically using Gaussian noise, and then minimize the distance between the reconstruction of $\mathbf{x}'$ and the *un-corrupted* image $\mathbf{x}$. The motivation is to learn representations that are insensitive to the added noise. Our work differs in that we do not constrain the decoder to recover the original image from the corrupted image but, rather, to constrain the encoder to recover the latent representation of the original image from the corrupted image via a KL divergence minimization constraint.

Contractive auto-encoders (CAES) (Rifai et al., 2011) share a similar goal with CR-VAES. A CAE is an auto-encoder whose encoder is constrained by minimizing the norm of the Jacobian of the output of the encoder with respect to the input image. This norm constraint on the Jacobian forces the representations learned by the encoder to be insensitive to changes in the input. Our work differs in several main ways. First, CR-VAES are not deterministic auto-encoders, contrary to CAES. We can easily sample from a CR-VAE, as for any VAE, which is not the case for a CAE. Second, a CAE does

**Table 1:** CR-VAES learn better representations than their base VAES on all three benchmark datasets. Although fitting the base VAE with augmentations does improve the representations, adding the consistency regularization further improves the quality of these learned representations. The value of $\beta$ for the $\beta$-VAE is inside the parentheses.

| Method | MNIST | | OMNIGLOT | | CELEBA | |
|---|---|---|---|---|---|---|
| | MI | AU | MI | AU | MI | AU |
| VAE | $124.5 \pm 1.1$ | $36 \pm 0.8$ | $105.4 \pm 1.2$ | $50 \pm 0.0$ | $33.8 \pm 0.2$ | $32 \pm 0.9$ |
| VAE + Aug | $125.9 \pm 0.2$ | $42 \pm 0.5$ | $105.9 \pm 0.7$ | $50 \pm 0.0$ | $34.1 \pm 0.8$ | $33 \pm 0.9$ |
| CR-VAE | $\mathbf{126.3 \pm 0.9}$ | $\mathbf{47 \pm 0.5}$ | $\mathbf{107.8 \pm 1.1}$ | $50 \pm 0.0$ | $\mathbf{34.9 \pm 0.5}$ | $\mathbf{33 \pm 1.2}$ |
| IWAE | $127.1 \pm 0.7$ | $39 \pm 0.5$ | $110.3 \pm 1.1$ | $50 \pm 0.0$ | $36.9 \pm 0.5$ | $36 \pm 1.6$ |
| IWAE+Aug | $129.0 \pm 0.9$ | $45 \pm 0.8$ | $112.9 \pm 0.7$ | $50 \pm 0.0$ | $37.0 \pm 0.2$ | $36 \pm 1.2$ |
| CR-IWAE | $\mathbf{129.7 \pm 1.0}$ | $\mathbf{50 \pm 0.0}$ | $\mathbf{115.3 \pm 0.8}$ | $50 \pm 0.0$ | $\mathbf{38.4 \pm 0.5}$ | $36 \pm 1.9$ |
| $\beta$-VAE (0.5) | $284.3 \pm 1.1$ | $50 \pm 0.0$ | $143.4 \pm 1.0$ | $50 \pm 0.0$ | $75.8 \pm 0.5$ | $49 \pm 0.5$ |
| $\beta$-VAE (0.5) + Aug | $289.3 \pm 1.0$ | $50 \pm 0.0$ | $159.6 \pm 1.3$ | $50 \pm 0.0$ | $75.7 \pm 0.3$ | $49 \pm 0.0$ |
| $\beta$-CR-VAE (0.5) | $\mathbf{291.9 \pm 0.7}$ | $50 \pm 0.0$ | $\mathbf{169.5 \pm 0.5}$ | $50 \pm 0.0$ | $\mathbf{77.1 \pm 0.1}$ | $\mathbf{50 \pm 0.0}$ |
| $\beta$-VAE (10) | $6.3 \pm 0.6$ | $8 \pm 1.7$ | $1.4 \pm 0.2$ | $4 \pm 0.9$ | $3.6 \pm 0.3$ | $7 \pm 0.8$ |
| $\beta$-VAE (10) + Aug | $6.5 \pm 0.5$ | $9 \pm 1.1$ | $1.6 \pm 0.2$ | $4 \pm 0.5$ | $3.7 \pm 0.1$ | $7 \pm 0.0$ |
| $\beta$-CR-VAE (10) | $\mathbf{6.9 \pm 0.6}$ | $\mathbf{10 \pm 0.5}$ | $\mathbf{1.6 \pm 0.1}$ | $\mathbf{4 \pm 0.5}$ | $\mathbf{3.7 \pm 0.4}$ | $\mathbf{9 \pm 0.9}$ |

not apply transformations to the input image, which limits the sensitivities it can learn to limit to those exhibited in the training set. Finally, CAES use the Jacobian to impose a consistency constraint, which are not as easy to compute as the KL divergence we use on the variational distribution induced by the encoder.

# 4 Empirical Study

In this section we show that a CR-VAE improves the learned representations of its base VAE and positively affects generalization performance We also show that the proposed regularization method is amenable to different VAE variants by applying it not only to the original VAE but also to the IWAE, the $\beta$-VAE, and the NVAE. We showcase the importance of the KL regularization term by conducting an ablation study. We found that only regularizing with data augmentation improves performance but that accounting for the KL term ($\lambda > 0$) further improves the quality of the learned representations and generalization.

We will conduct three sets of experiments. In the first experiment, we will apply the regularization method proposed in this paper to standard VAES such as the original VAE, the IWAE, and the $\beta$-VAE. We use MNIST, OMNIGLOT, and CELEBA as datasets for this experiment. For CELEBA, we choose the 32x32 resolution for this experiment. Our results show that adding consistency regularization always improves upon the base VAE, both in terms of the quality of the learned representations and generalization. We conduct an ablation study and also report performance of the different VAE variants above when they are fitted with the original data and their augmentations. The results from this ablation highlight the importance of setting $\lambda > 0$.

In the second set of experiments we apply our method to a large-scale VAE, the latest NVAE (Vahdat & Kautz, 2020). We use MNIST, CIFAR-10, and CELEBA as datasets for this experiment. We increased the resolution for the CELEBA dataset for this experiment to 64x64. We reach the same conclusions as for the first sets of experiments; CR-VAES improve the learned representations and generalization of their base VAES. In this particular setting, the CR-NVAE achieves state-of-the-art generalization performance on both MNIST and CIFAR-10. This state-of-the-art performance couldn't be reach simply by training the NVAE with augmentations, as our results show.

Finally, in a third set of experiments, we apply our regularization technique to a 3D point-cloud dataset called ShapeNet (Chang et al., 2015). We adapt a high-performing auto-encoding method called FoldingNet (Yang et al., 2018) to its VAE counterpart and apply the method we described in this paper to that VAE variant on the ShapeNet dataset. We found that adding consistency regularization yields better learned representations.

We next describe in great detail the set up for each of these experiments and the results showcasing the usefulness of the regularization method we propose in this paper.

**Table 2:** CR-VAES learn representations that yield higher accuracy on downstream classification than their base VAES. These results correspond to the accuracy from a linear classifier that was fitted on the training. We fed this classifier with the representations learned by each method. On both MNIST and CIFAR-10, CR-VAES yield higher accuracy.

| Method | MNIST | CIFAR-10 |
|---|---|---|
| VAE | 98.5 | 32.6 |
| VAE+Aug | 98.9 | 40.1 |
| CR-VAE | **99.4** | **44.7** |
| IWAE | 98.6 | 35.8 |
| IWAE+Aug | **99.9** | 37.1 |
| CR-IWAE | **99.9** | **44.8** |
| $\beta$- VAE (0.5) | 97.6 | 27.0 |
| $\beta$- VAE (0.5)+Aug | 98.7 | 27.6 |
| $\beta$- CR-VAE (0.5) | **98.9** | **30.0** |
| $\beta$- VAE (10) | 99.4 | 36.5 |
| $\beta$- VAE (10)+Aug | **99.6** | 42.1 |
| $\beta$- CR-VAE (10) | **99.6** | **46.1** |

**Table 3:** CR-VAES generalize better than their base VAES on almost all cases; they achieve lower negative log-likelihoods. Although training the base VAES with the augmented data improves generalization, adding the consistency regularization term further improves generalization performance.

| Method | MNIST | OMNIGLOT | CELEBA |
|---|---|---|---|
| VAE | $83.7 \pm 0.3$ | $128.2 \pm 0.8$ | $66.1 \pm 0.2$ |
| VAE + Aug | $82.8 \pm 0.4$ | $125.7 \pm 0.2$ | $66.0 \pm 0.2$ |
| CR-VAE | $\mathbf{81.2 \pm 0.2}$ | $\mathbf{124.1 \pm 0.1}$ | $\mathbf{65.9 \pm 0.2}$ |
| IWAE | $81.7 \pm 0.3$ | $127.5 \pm 0.5$ | $65.3 \pm 0.1$ |
| IWAE+Aug | $80.4 \pm 0.2$ | $125.0 \pm 0.6$ | $65.3 \pm 0.1$ |
| CR-IWAE | $\mathbf{79.7 \pm 0.3}$ | $\mathbf{123.6 \pm 0.5}$ | $\mathbf{65.0 \pm 0.2}$ |
| $\beta$-VAE (0.5) | $92.6 \pm 0.3$ | $137.1 \pm 0.2$ | $68.7 \pm 0.2$ |
| $\beta$-VAE (0.5) + Aug | $90.0 \pm 0.5$ | $134.6 \pm 0.5$ | $68.8 \pm 0.2$ |
| $\beta$-CR-VAE (0.5) | $\mathbf{85.7 \pm 0.6}$ | $\mathbf{132.5 \pm 0.3}$ | $\mathbf{68.2 \pm 0.1}$ |
| $\beta$-VAE (10) | $\mathbf{126.1 \pm 1.8}$ | $157.5 \pm 1.1$ | $92.7 \pm 0.5$ |
| $\beta$-VAE (10) + Aug | $127.1 \pm 1.0$ | $\mathbf{157.3 \pm 0.5}$ | $92.7 \pm 0.3$ |
| $\beta$-CR-VAE (10) | $126.2 \pm 0.5$ | $157.6 \pm 0.6$ | $\mathbf{92.6 \pm 0.1}$ |

### 4.1 Application to standard VAES on benchmark datasets

We apply consistency regularization, as described in this paper, to the original VAE, the IWAE, and the $\beta$-VAE. We now describe the set up and results for this experiment.

**Datasets.** We study three benchmark datasets that we briefly describe below. We first consider MNIST. MNIST is a handwritten digit recognition dataset with $60,000$ images in the training set and $10,000$ images in the test set (LeCun, 1998). We form a validation set of $10,000$ images randomly sampled from the training set.

We also consider OMNIGLOT, a handwritten alphabet recognition dataset (Lake et al., 2011). This dataset is composed of $19,280$ images. We use $16,280$ randomly sampled images for training and $1,000$ for validation and the remaining $2,000$ samples for testing.

Finally we consider CELEBA. It is a dataset of faces, consisting of $162,770$ images for training, $19,867$ images for validation, and $19,962$ images for testing (Liu et al., 2018). We set the resolution to 32x32 for this experiment.

**Transformations** $t(\tilde{\mathbf{x}}|\mathbf{x})$**.** We consider three transformations variants for image data $t(\tilde{\mathbf{x}}|\mathbf{x})$. The first randomly translates an image $[-2, 2]$ pixels in any direction. The second transformation randomly

**Table 4:** The regularization term $\lambda$ affects both generalization performance and the quality of the learned representations. Many values of $\lambda$ perform better than the base VAE. However a large enough value of $\lambda$, e.g. $\lambda = 1$, can lead to worse performance than the base VAE because for large values of $\lambda$ the regularization term takes over the data-term in the objective function.

|  | $\lambda$ | MI | AU | NLL |
|---|---|---|---|---|
| VAE | $--$ | 124.5 | 36 | 83.7 |
| CR-VAE | 0.001 | 125.0 | 38 | 83.5 |
| CR-VAE | 0.01 | 125.9 | 41 | 82.4 |
| **CR-VAE** | **0.1** | **126.3** | **47** | **81.2** |
| CR-VAE | 1 | 124.3 | 47 | 83.9 |

**Table 5:** The choice of augmentation affects both generalization performance and the quality of the learned representations. Jointly using all augmentations works best.

| Augmentation | MI | AU | NLL |
|---|---|---|---|
| Rotations only | 125.8 | 45 | 82.1 |
| Translations only | 126.1 | 45 | 81.9 |
| Scaling only | 125.1 | 42 | 82.7 |
| All | **126.3** | **47** | **81.2** |

rotates an image uniformly in $[-15, 15]$ degrees clockwise. Finally the third transformation randomly scales an image by a factor uniformly sampled from $[0.9, 1.1]$.

**Evaluation metrics.** The regularization method we propose in this paper is mainly aimed at improving the learned representations of VAEs. To assess these representations we use three metrics: mutual information, number of active latent units, and accuracy on a downstream classification task. We also evaluate the effect of the proposed method on generalization to unseen data. For that we also report negative log-likelihood. We define each of these metrics next.

*Mutual information (MI).* The first quality metric is the mutual information $I(\mathbf{z}; \mathbf{x})$ between the observations and the latents under the joint distribution induced by the encoder,

$$I(\mathbf{z}; \mathbf{x}) = \mathbb{E}_{p_d(\mathbf{x})} \left[ KL(q_\phi(\mathbf{z}|\mathbf{x})||p(\mathbf{z})) - \text{KL}(q_\phi(\mathbf{z})||p(\mathbf{z})) \right] \tag{5}$$

where $p_d(\mathbf{x})$ is the empirical data distribution and $q_\phi(\mathbf{z})$ is the *aggregated posterior*, the marginal over $\mathbf{z}$ induced by the joint distribution defined by $p_d(\mathbf{x})$ and $q_\phi(\mathbf{z}|\mathbf{x})$. The mutual information is intractable but we can approximate it with Monte Carlo. Higher mutual information corresponds to more interpretable latent variables.

*Number of active latent units (AU).* The second quality metrics we consider is the number of active latent units (AU). It is defined in Burda et al. (2015) and measures the "activity" of a dimension of the latent variables $\mathbf{z}$. A latent dimension is "active" if

$$Cov_{\mathbf{x}}(\mathbb{E}_{\mathbf{u} \sim q_\phi(\mathbf{u}|\mathbf{x})}) > \delta \tag{6}$$

where $\delta$ is a threshold defined by the user. For our experiments we set $\delta = 0.01$. The higher the number of latent active units, the better the learned representations.

*Accuracy on downstream classification.* This metric is calculated by fitting a given VAE, taking the learned representations for each data in the test set and computing the accuracy from the prediction of the labels of the images in that same test set by a classifier fitted on the training set. This metric is only applicable to labelled datasets.

*Negative log-likelihood.* We use negative held-out log-likelihood to assess generalization. Consider an unseen data $\mathbf{x}^*$, its negative held-out log-likelihood under the fitted model is

$$\log p_\theta(\mathbf{x}^*) = -\log \left( \mathbb{E}_{q_\phi(\mathbf{z}|\mathbf{x}^*)} \left[ \frac{p_\theta(\mathbf{x}^*, \mathbf{z})}{q_\phi(\mathbf{z}|\mathbf{x}^*)} \right] \right). \tag{7}$$

**Table 6:** The CR-VAE outperforms a popular and advanced contrastive learning technique called *triplet loss* on both generalization performance and quality of learned representations.

| Method | MI | AU | NLL |
|---|---|---|---|
| VAE | 124.5 | 36 | 83.7 |
| VAE + augmentations | 125.9 | 42 | 82.8 |
| VAE + triplet loss | 124.9 | 39 | 83.1 |
| CR-VAE | **126.3** | **47** | **81.2** |

This is intractable and we approximate it using Monte Carlo,

$$\log p_\theta(\mathbf{x}^*) \approx -\log \frac{1}{S} \sum_{s=1}^{S} \frac{p_\theta(\mathbf{x}^*, \mathbf{z}^{(s)})}{q_\phi(\mathbf{z}^{(s)}|\mathbf{x}^*)} \tag{8}$$

where $\mathbf{z}^{(1)}, \ldots, \mathbf{z}^{(S)} \sim q_\phi(\mathbf{z}|\mathbf{x}^*)$.

**Settings.** The VAES are built on the same architecture as Tolstikhin et al. (2017). The networks are trained with the Adam optimizer with a learning rate of $10^{-4}$ (Kingma & Ba, 2014) and trained for 100 epochs with a batch size of 64. We set the dimensionality of the latent variables to 50, therefore the maximum number of active latent units in the latent space is 50. We found $\lambda = 0.1$ to be best according to cross-validation using held-out log-likelihood and exploring the range $[1e^{-4}, 1.0]$ datasets. In an ablation study we explore $\lambda = 0$. For the $\beta$-VAE we set $\lambda = 0.1 \cdot \beta$ and study both $\beta = 0.1$ and $\beta = 10$, two regimes under which the $\beta$-VAE performs qualitatively very differently (Higgins et al., 2017). All experiments were done on a GPU cluster consisting of Nvidia P100 and RTX. The training took approximately 1 day for most experiments.

**Results.** Table 1 shows that on all the three benchmark datasets all the different VAE variants we studied, consistency regularization as developed in this paper always improves the quality of the learned representations as measured by mutual information and the number of active latent units. These results are confirmed by the numbers shown in Table 2 where CR-VAES always lead to better accuracy on downstream classification.

We proposed consistency regularization as a way to improve the quality of the learned representations. Incidentally, Table 3 also shows that it can improve generalization as measured by negative log-likelihood.

**Ablation Study.** We now look at the impact of each factor that goes into the regularization method we introduced in this paper using MNIST. We test the impact of the regularization term $\lambda$ and the impact of the choice of augmentation on all metrics. Table 4 and Table 5 show the results.

Table 4 shows that even small consistency regularization (a small $\lambda$ value) results in improvement over the base VAE but that a large enough $\lambda$ value can hurt performance.

Table 5 shows that rotations and translations are more important than scaling, but the combination of all three augmentations works best for CR-VAES.

**Comparison to Contrastive Learning.** We look at how CR-VAES compare against a popular and advanced contrastive-learning-based technique, the *triplet loss* (Schroff et al., 2015) using MNIST. Table 6 shows that the CR-VAE outperforms the triplet loss on both generalization performance and quality of learned representations. Table 6 also confirms existing literature showing simply applying augmentations can outperform complex contrastive learning-based methods such as the triplet loss (Kostrikov et al., 2020; Sinha & Garg, 2021).

### 4.2 Application to the large-scale NVAE on benchmark datasets

Along with standard VAE variants, we also experiment with a large scale state-of-the-art VAE, the NVAE(Vahdat & Kautz, 2020). Similar to before, we simply add consistency regularization using the image-based augmentations techniques to the NVAE model and experiment on benchmark datasets: MNIST (LeCun, 1998), CIFAR-10 (Krizhevsky et al., 2009) and CELEBA (Liu et al., 2018).

The results for large scale generative modeling are tabulated in Table 8 and Table 7, where we see that using CR-NVAE we are able to learn representations that yield better accuracy on downstream

**Table 7:** The CR-NVAES learns better representations than the base NVAE as measured by accuracy on a downstream classification on both MNIST and CIFAR-10. We get to this same conclusion when looking at the number of active units as an indicator for the quality of the learned latent representations; CR-NVAE recovers 226 units whereas NVAE recovers 211 units.

| Method | MNIST | CIFAR-10 |
|---|---|---|
| NVAE | **99.9** | 57.9 |
| NVAE+Aug | **99.9** | 66.4 |
| CR-NVAE | **99.9** | **71.4** |

**Table 8:** Large-scale experiments with NVAES with and without consistency-regularization on 3 benchmark datasets: dynamically binarized MNIST, CIFAR-10 and CELEBA. We report generalization using negative log-likelihood on MNIST and bits per dim on CIFAR-10 and CELEBA. On all datasets consistency regularization improves generalization performance. In particular CR-NVAE achieves state-of-the-art performance on MNIST and CIFAR-10.

| | MNIST ($28 \times 28$) | CIFAR-10 ($32 \times 32$) | CELEBA ($64 \times 64$) |
|---|---|---|---|
| NVAE | 78.19 | 2.91 | 2.03 |
| NVAE+Aug | 77.53 | 2.70 | 1.96 |
| CR-NVAE | **76.93** | **2.51** | **1.86** |

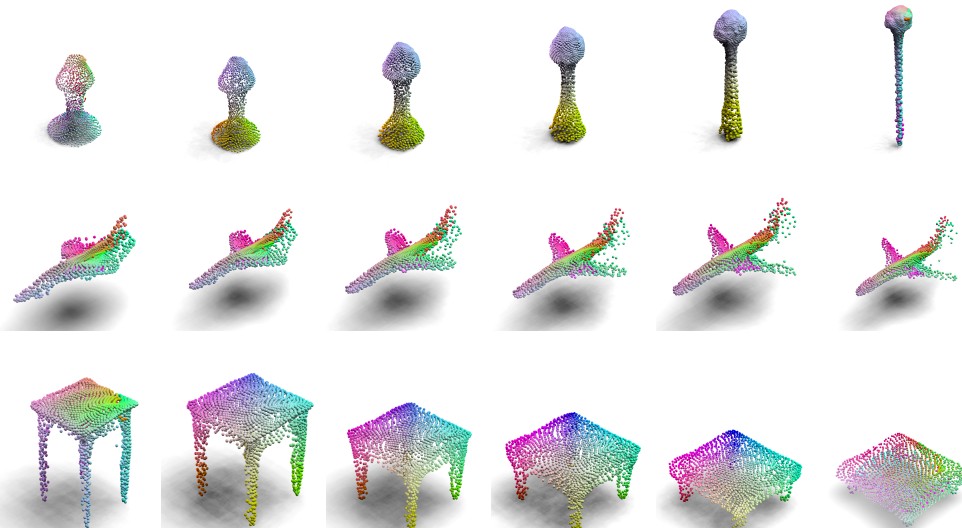

**Figure 2:** Interpolation between two samples of a lamp, airplane and table using a trained CR-FoldingNet trained on the ShapeNet dataset. The CR-FoldingNet is able to learn an interpretable latent space.

classification and set new state-of-the-art values on each of the datasets, improving upon the baseline log-likelihood values. This shows the ability of consistency regularization to work at scale on challenging generative modeling tasks.

### 4.3 Application to the FoldingNet on 3D point-cloud data

Along with working with image data, we additionally experiment with 3D point cloud data using a FoldingNet Yang et al. (2018) and the ShapeNet dataset Chang et al. (2015) which consists of 55 distinct object classes. FoldingNet learns a deep AutoEncoder to learn unsupervised representations from the point cloud data. To add consistency regularization, we first substitute the AutoEncoder to a

**Table 9:** The FoldingNet yields higher accuracy when paired with consistency regularization on the ShapeNet dataset. The results shown here correspond to a FoldingNet that was trained with augmented data, the same used to apply consistency regularization. As can be seen from these results, enforcing consistency through KL as we do in this paper leads to representations that perform well on a downstream classification. Here the classifier used is a linear SVM. We also report mean reconstruction error through Chamfer distance where the same conclusion holds.

| Method | Accuracy | Reconstruction Loss |
|---|---|---|
| Folding Net (Aug) | 82.5% | 0.0355 |
| CR-Folding Net | **84.6%** | **0.0327** |

VAE by adding the KL term from the ELBO to the baseline FoldingNet. We then add the additional consistency regularization KL term to the latent space of FoldingNet.

For the ShapeNet point cloud data, we perform data augmentation using a similar scheme to what we did for the previous experiments, we randomly translate, rotate and add jitter to the $(x, y, z)$ coordinates of the point cloud data. We follow the same scheme detailed in FoldingNet (Yang et al., 2018).

We train both the FoldingNet turned in a VAE and the CR-FoldingNet with these augmentations. To train CR-FoldingNet, we additionally apply the consistency regularization term as proposed in Equation 3. The results on the validation set for reconstruction (as measured by Chamfer distance) and accuracy are shown in Table 9.

We also visualize the point clouds reconstructions and interpolations between 3 different object classes using a CR-FoldingNet in Figure 2. We perform 4 interpolation steps for each of the objects, to highlight the interpretable learned latent space. Additionally, we perform the same interpolation on the baseline FoldingNet model. We show these interpolations in the appendix.

## 5    Conclusion

We proposed a simple regularization technique to constrain encoders of VAES to learn similar latent representations for an image and a semantics-preserving transformation of the image. The idea consists in maximizing the likelihood of the pair of images while minimizing the KL divergence between the variational distribution induced by the encoder when conditioning on the image on one hand, and its transformation, on the other hand. We applied this technique to several VAE variants on several datasets, including a 3D dataset. We found it always leads to better learned representations and also better generalization to unseen data. In particular, when applied to the NVAE, the regularization technique we developed in this paper yields state-of-the-art results on MNIST and CIFAR-10.

## Broader Impact

In this paper, we propose a simple method that performs a KL-based consistency regularization scheme using data augmentation for VAES. The broader impact of the study includes practical applications such as graphics and computer vision applications. The method we propose improves the learned representations of VAES, and as an artifact, also improves their generalization to unseen data. In this regard, any implications of VAES also apply to this work. For example, the generative model fit by a VAE may be used to generate artificial data such as images, text, and 3D objects. Biases may arise as a result of poor data selection. Furthermore, text generated from generative systems may amplify harmful speech contained in the data. However, the method we propose can also improve the performance of VAES when used in certain practical domains as we discussed in the introduction of the paper.

## 6    Acknowledgements

We thank Kevin Murphy, Ben Poole, and Augustus Odena for their comments on this work.

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
