# OpenReview forum: "Consistency Regularization for Variational Auto-Encoders"
_NeurIPS.cc/2021/Conference — NeurIPS 2021 Poster_

### Official Review · Reviewer_69Xg · 2021-06-28

**Rating:** 6
**Confidence:** 5

**Summary:**

This paper proposes a study on enforcing semantic consistency on VAEs. The authors show that the current VAEs are sensitive to semantical-preserving transformations and it will lower the quality of representations and generalization. To solve this issue, the authors propose a new algorithm, which first transforms a training sample into multiple semantically similar samples and then minimize the KL-divergence between the encoded distributions of original and new samples. Experiments on both 2D and 3D datasets show that the new algorithm is effective in increasing representation quality of VAEs.

**Limitations And Societal Impact:**

The authors mention that this paper might be used to generate artificial data or introduce biases. However, I think these are common problems for all deep generative models. So I don't think the authors need to improve this specific paper to solve the potential negative impacts.

**Main Review:**

Pros:
1. This paper discusses a very interesting problem, how to make representations semantically invariant.
2. The method is simple and can be applied to all variants of VAEs.
3. Experiments show the proposed algorithm is useful in a wide range of datasets.

Cons:
1. This paper looks like a direct application of contrastive learning on VAEs. Though the loss function is new,  the idea, algorithm, and even transformations are very similar to works on contrastive learning. The authors mention CAEs in line 130. However, more advanced contrastive methods are not compared as baselines.
2. Some experiment results are not significant.
  (a) In Table2, the accuracy increase of using CR on MNIST is marginal.
  (b) In Table 3, the differences of estimated log-likelihoods are also small in most settings.
3. The choice of some parameters is not verified.
  (a) The most important parameter \lambda. I cannot figure out how to choose it. The authors say that ablation studies are performed, but I still cannot find the results.
  (b) The transformations (including translation, rotation, and scaling) and their parameters fully depend on handcrafting. Though it is a common problem for contrative learning, the authors should think about how to do it more automatically.

Other comments:
1. A large portion of this paper is devoted to introducing variational autoencoders and other background knowledge. For example, the first several paragraphs in the introduction and method part. The authors should talk more about their own staff with a very brief introduction of necessary backgrounds.

**Time Spent Reviewing:**

4

---

> ### Author Response · Authors · 2021-08-10
> **Thank you for the review!**
>
> Thank you for the review! We will respond to each of your points directly in the comments below.
>
> **Insignificant improvement gains for MNIST in table 2 and log-likelihood improvements in table 3**
> - For Table 2, we report the accuracy for the MNIST dataset, which may seem insignificant, since MNIST is easily separable in the latent space for vanilla VAEs. The error rate for MNIST can instead be used to show how we improve the error rate for MNIST from 1.5% to 0.6% for the vanilla VAEs. The experiments with CIFAR-10 results are more significant as the data distribution is harder to learn for vanilla VAEs.
> - The NLL results in table 3 for example of ~2.5 using CR-VAEs trained on MNIST is in fact a significant improvement, as even small NLL improvements are hard to achieve. For example IWAE [1] shows an improvement of ~2.0 on the NLL metric on MNIST over a vanilla VAE (see table 1 of Burda et al. [1]).
>
> **Ablation with \lambda**
> - Thank you for the suggestion; ablating over \lambda is indeed an important study since it is the main hyperparameter for CR-VAEs. Here is the experiment as suggested for the MNIST dataset. We will add these results to the final draft. We see that a \lambda value that is too small results in improvement over the baseline, however a \lambda value that is too high can hurt performance slightly.
> |                        | MI        | AU     | NLL      |
> |------------------------|-----------|--------|----------|
> | VAE                    | 124.5     | 36     | 83.7     |
> | CR-VAE \lambda = 0.001 | 125.0     | 38     | 83.5     |
> | CR-VAE \lambda = 0.01  | 125.9     | 41     | 82.4     |
> | CR-VAE \lambda = 0.1   | **126.3** | **47** | **81.2** |
> | CR-VAE \lambda = 1     | 124.3     | **47** | 83.9     |
>
> **Augmentations are handcrafted**
> - In computer vision research, spatial augmentations (such as rotations, translation and scaling) and colour augmentations (such as colour jitters) are commonly used, since they preserve the semantics of the original image. Similar semantic-preserving augmentations exist for other modalities of data, such as local deformations for 3D point clouds, which can be easily added to the CR-VAE framework, as we show using CR-FoldingNets in Section 4.3. Furthermore, we use the same augmentation scheme for all of the image based experiments, which suggests that the augmentation strategies are not handcraft and can generalize to other image datasets. We also show experiments with MNIST and CIFAR, which are different data distributions, as one is grayscale and our other experiments use natural images. Using the same data augmentations for both data distributions also suggests that the augmentations are in fact not handcrafted, and can generalize well to a wide range of image based datasets.
>
>
> [1] https://arxiv.org/abs/1509.00519

---

> > ### Comment · Reviewer_69Xg · 2021-08-16
> > **Further comments**
> >
> > Thank you for your responses! But could you please answer my main concern (point 1 in Cons)? If I'm convinced by your answer, I will change my rating.

---

> > > ### Author Response · Authors · 2021-08-17
> > > **Thank you for your time to check the response**
> > >
> > > Thank you for taking the time and reading our rebuttal, and apologies for missing con #1. We have addressed it in this comment.
> > >
> > >  It is true that there are more advanced contrastive learning techniques in the contrastive learning literature (triplet losses for example), but they are yet to be applied successfully for VAEs. One baseline that we do consider is training the model with data augmentation only (\lambda = 0), which has been shown to  outperform models trained with more complicated contrastive learning techniques [1, 2].
> > >
> > > For completeness, we trained the vanilla VAE baseline on MNIST using a triplet loss in the learned embedding space. The results are as follows:
> > >
> > > |                       | MI        | AU     | NLL      |
> > > |-----------------------|-----------|--------|----------|
> > > | VAE                   | 124.5     | 36     | 83.7     |
> > > | VAE+triplet loss      | 124.9     | 39     | 83.1     |
> > > | VAE+only augmentation | 125.9     | 42     | 82.8     |
> > > | CR-VAE                | **126.3** | **47** | **81.2** |
> > >
> > > We can clearly see that CR-VAEs are able to outperform popular contrastive learning techniques such as a triplet loss and only using augmentations. We will add this discussion to the final draft.
> > >
> > > Please let us know if there are any remaining concerns regarding the review, and we would be happy to answer them.
> > >
> > >
> > >
> > > [1] https://arxiv.org/abs/2004.13649
> > >
> > > [2] https://arxiv.org/abs/2103.06326

---

> > > > ### Comment · Reviewer_69Xg · 2021-08-17
> > > > **Thanks for adding this comparison.**
> > > >
> > > > The result sounds good. So I will increase my score.

---

> > > > > ### Author Response · Authors · 2021-08-22
> > > > > **Thank you for the update**
> > > > >
> > > > > Are there any other problems we haven't addressed yet? What makes the paper still "marginally above acceptance threshold"? We would be glad to address any pending concerns you may have.
> > > > >
> > > > > Thank you again for the feedback.

---

### Official Review · Reviewer_aPrg · 2021-07-11

**Rating:** 7
**Confidence:** 4

**Summary:**

The paper proposes a novel method of using data augmentation when training Variational Autoencoders. Given a family of semantics-preserving transformations, the idea is to enforce the invariance of latent code w.r.t. these transformations by adding a KL penalty term. This penalty improves the quality of learned latent representation and of the generative model significantly.



**Limitations And Societal Impact:**

The paper could be further improved by considering different families of augmentations and analysing the impact of the augmentation choice on latent representation and log-likelihood.

Another interesting extension to the proposed method could be to assume equivariance of latent code rather then invariance, by parameterizing learnable(e.g. affine) transforms acting on z variable and enforcing the equivalence of  x -> z -> z' and x -> x' -> z'.



**Main Review:**

The paper proposes a novel method of using data augmentation when training Variational Autoencoders. Given a family of semantics-preserving transformations, the idea is to enforce the invariance of latent code w.r.t. these transformations by adding a penalty term. The penalty is KL divergence between encoder distribution at the augmented data point and encoder distribution at the original point.

Authors do extensive tests of the impact of this penalty on the quality of learned latent representations. Most experiments are performed on image datasets with augmentations being shifts, rotations and scaling. A variety of VAE architectures show improved quality of latents, compared to the standard way of applying augmentations. Quality of representations is accessed using mutual infromation estimates, active latent units and linear classifier accuracy (trained on latent features)
Authors also find that proposed method leads to better generative models with higher log-likelihood.

The proposed idea is original in the VAE context. The paper is well-written and experiments convincingly demonstrate the value of the proposed penalty.

Due to it's simplicity and effectiveness the proposed method will likely be widely used in practice.

*****************************
Post-rebuttal

I'd like to thank the authors for their response and will keep my already high score the same.

**Time Spent Reviewing:**

3

---

> ### Author Response · Authors · 2021-08-10
> **Thank you for the review!**
>
> Thank you for the review! We will respond to each of your points directly in the comments below.
>
> **Impact of augmentation choice**
> - Thank you for the suggestion. We have included the experiments of the effect of each of the augmentation individually on the MNIST dataset. It is clear to see that rotations and translations are more important than scaling, but the combination of all 3 works the best for CR-VAEs.
> |                   | MI        | AU     | NLL      |
> |-------------------|-----------|--------|----------|
> | Rotations only    | 125.8     | 45     | 82.1     |
> | Translations only | 126.1     | 45     | 81.9     |
> | Scaling only      | 125.1     | 42     | 82.7     |
> | All (CR-VAE)      | **126.3** | **47** | **81.2** |
>
> **Extension of the work regarding the equivariance properties of the model to latent-space augmentations**
> - Extending CR-VAEs to equivariance is indeed an interesting line of future work! We will add this discussion to the conclusion as a possible future extension to CR-VAEs.

---

### Official Review · Reviewer_svud · 2021-07-16

**Rating:** 6
**Confidence:** 5

**Summary:**

The paper improves VAEs by ensuring that the learned encoder maps augmented inputs $\tilde{x}$ and unaugmented inputs $x$ to the same representations. To do so, they minimize the KL divergence between the encoder conditioned on $\tilde{x}$ and $x$. They empirically show that such consistency regularised VAE (CR-VAE) achieves very strong density modeling results.

**Limitations And Societal Impact:**

As previously stated, the authors need to discuss the limitation of CR-VAE's representation compared to standard representation learning methods.

Potential societal impact is adequately discussed.

**Main Review:**

**Overall**: This is a clear paper that provides a very simple and useful method to improve VAE’s. My main issue is that the authors motivate the work as a way to learn useful representations but they (1) do not appropriately discuss related work in representation learning; (2) do not provide empirical evidence that supports the usefulness of the CR-VAE's representations compared to standard baselines in representation learning. Overall, I still think that the proposed method will be useful and interesting to people that work with VAEs.

**Strengths**
- **Very simple method**: the proposed method is very simple, which means that it can easily be applied to any VAE in a few lines of code.
- **Very clear paper**: the paper is very clear and well written.
- **Strong results for density modeling**: the log-likelihood results on MNIST / CIFAR / Celeba are strong, and the gains are significant compared to only using data augmentations.

**Weaknesses**
- **Missing related work**  this paper is essentially trying to learn representations that are invariant to standard augmentations. As a result, this makes the proposed method very similar to self-supervised learning (SSL). For example, the same regulariser already appeared in [1].  This decreases somewhat the novelty of CR-VAE as these ideas have been well investigated for representation learning. From my perspective, this paper uses ideas from SSL to improve VAE’s. This is not inherently an issue as long as SSL is correctly discussed (note that lines 119-121 won’t be true anymore). Specifically:
    - [1]is very related to your method and uses the same regulariser. The key difference is that instead of reconstructing the image (learning p(x|z)) they use contrastive learning to maximize the mutual information I[Z, X]. This exemplifies well the relation between your method and SSL.
    - Most SSL methods aim to make representations of an augmented and unaugmented example very similar. For example [2] does that by minimizing the variance instead of KL divergence (see their Appendix A for relation to KL divergence).  Similar ideas of learning approximately invariant representations go way back, e.g., [3] from 2006.
    - [4] seems to be another very related work (although online since the beginning of March so contemporary). In particular, the Variational Invariant Compressor (sec. 4.1.) is essentially a VAE that takes as input augmented inputs but reconstructs unaugmented inputs to ensure “consistency” of the encoder.
- **Lack of results showing the usefulness of the learned representations**: an important motivation of the work seems to be about learning useful representations, yet empirical results do not really support this claim:
    - the experimental results of downstream linear classifications are extremely weak compared to typical SSL methods, which suggests that the learned representations of CR-VAE are not useful for downstream tasks. Indeed, the best CIFAR-10 accuracy in this paper is 71.4% while standard SSL with linear evaluation achieves more than 94% (e.g. SimCLR Appendix B.9.).
    - The authors evaluate representation learning using mutual information, but (1) they do not compare to typical representation learning baselines; (2) it is well known that higher mutual information does not mean better representations [5].
- **Somewhat incomplete experimental results**: Given that this is a completely empirical paper (based on sensible intuition) I find the experimental section to be somewhat incomplete. Here are results that I would have liked to see:
    - (As previously said) a result that shows the usefulness of representations
    - Lineplot showing for one dataset (e.g. CIFAR10) the effect of $\lambda$ on the log-likelihood of CR-VAE, that would help to understand the robustness of the model to the additional hyperparameter.
    - Experiment analyzing the impact of the choice of augmentations on CR-VAE, to understand the importance of the choice of augmentations.
    - Standard errors for table 2/4/5/6. Especially table 5 as you claim SOTA.
    - Qualitative samples from a CR-VAE for 2D images to understand the effect of the regulariser.

**Suggestions / specific issues**
- I’m not fond of your use of “consistency”, which means something very specific in (frequentist) statistics and the title could thus be understood as a method to ensure that the variational posterior converges to the true posterior (similar to [6]). I would at least consider using another word. I understand if the authors ultimately keep it given that it’s (unfortunately) the term used in semi-supervised learning.
- The paper [7] is in the references but never actually cited in the text. Given that this paper shows comparable results on CIFAR10 and really shows the importance of data augmentation for density modeling, I think that it should be correctly discussed in your related work section.
- Line 202 please explain in the text why more active units should be better

**Questions**
- Why did you choose the KL divergence in this direction? Please explain in the text, and ideally add an experimental comparison.
- Another way of forcing “consistency” of the representations would be to augment the input but reconstruct the unaugmented examples (as in [4]), Have you considered this instead of your regulariser? I’d be curious to hear what you think, although I wouldn’t be surprised if “unaugmenting” the inputs is harder to do in practice.

**Minor Suggestions**
- Line 18: state of the art performance -> state of the art log-likelihoods          (I initially thought that SOTA performance was on downstream classification)
- figure 1: I would add the sub captions for those that skim the paper: (a) VAE   (b) VAE with augmentations   (c) CR-VAE (ours)
- Line 71 / 91 : e.g. -> e.g.,
- Line 82: is unable -> may be unable
- Line 148: will conduct -> conduct
- Line 148 will apply -> apply
- Line 185: Finally -> Finally,
- Line 191: likelihood -> likelihoods
- Equation 5 : KLs use different fonts
- Line 198: metrics -> metric
- Line 199/200: using different quotes on left and right of “active”

**References**
- [1] Federici, Marco, et al. "Learning robust representations via multi-view information bottleneck." arXiv preprint arXiv:2002.07017 (2020).
- [2] Zbontar, Jure, et al. "Barlow twins: self-supervised learning via redundancy reduction." arXiv preprint arXiv:2103.03230 (2021).
- [3] Hadsell, Raia, Sumit Chopra, and Yann LeCun. "Dimensionality reduction by learning an invariant mapping." 2006 IEEE Computer Society Conference on Computer Vision and Pattern Recognition (CVPR'06). Vol. 2. IEEE, 2006.
- [4] Dubois, Yann, et al. "Lossy Compression for Lossless Prediction." arXiv preprint arXiv:2106.10800 (2021).
- [5] Tschannen, Michael, et al. "On mutual information maximization for representation learning." arXiv preprint arXiv:1907.13625 (2019).
- [6] Wang, Yixin, and David M. Blei. "Frequentist consistency of variational Bayes." Journal of the American Statistical Association 114.527 (2019): 1147-1161.
- [7] Jun, Heewoo, et al. "Distribution augmentation for generative modeling." International Conference on Machine Learning. PMLR, 2020.

**Time Spent Reviewing:**

6 hours

---

> ### Author Response · Authors · 2021-08-10
> **Thank you for the review!**
>
> Thank you for the review! We will respond to each of your points directly in the comments below.
>
> **Related work, writing fixes, and adding discussion regarding the active units metric**
> - Thank you for the relevant work and suggested edits. We will be sure to add the citations in the final version of the paper. We will also add a discussion about each of the chosen metrics (including active units), as suggested by the reviewer.
>
> **Comparison to Self-Supervised Learning literature like SimCLR**
> - Self-supervised learning has shown very impressive gains in learning representations. VAEs are another framework for learning representations. This work is focused on showing that the representations learned by any VAE can be improved by adopting the method we propose in the paper. Along with improving generative models, improving VAEs fundamentally can improve model performance on many different methods such as anomaly detection [1,2], language modeling [3], among others. In this paper, we are focused on improving VAEs in general.
>
> **Adding comments about how better MI does not necessarily mean better representations**
> - It is true that Mutual information is not a perfect measure of the quality of learned representations, which is why to demonstrate the effectiveness of CR-VAEs, we measure multiple metrics such as NLL, Active units, bits per dimension, reconstruction loss (with 3D point cloud data), downstream classification accuracy, alongside with mutual information. Our results show CR-VAEs improve upon their base VAE counterparts on all those metrics. The results on downstream classification further show evidence that CR-VAEs learn more predictive representations.
>
> **MNIST/CIFAR downstream classification not enough**
> - Alongside MNIST and CIFAR downstream classification experiments, we also perform downstream classification experiments with 3D point cloud data using the ShapeNet dataset. We are limited by compute to perform CR-NVAE experiments on a large scale dataset, such as ImageNet (as done by Vahdat et al.), since those experiments require 32 V-100 GPUs that are not accessible to the authors.
>
> **Ablations with different values of \lambda**
> - Thank you for the suggestion; ablating over \lambda is indeed an important study since it is the main hyperparameter for CR-VAEs. Here is the experiment as suggested for the MNIST dataset. We will add these results to the final draft. We see that a \lambda value that is too small results in improvement over the baseline, however a \lambda value that is too high can hurt performance slightly.
> |                        | MI        | AU     | NLL      |
> |------------------------|-----------|--------|----------|
> | VAE                    | 124.5     | 36     | 83.7     |
> | CR-VAE \lambda = 0.001 | 125.0     | 38     | 83.5     |
> | CR-VAE \lambda = 0.01  | 125.9     | 41     | 82.4     |
> | CR-VAE \lambda = 0.1   | **126.3** | **47** | **81.2** |
> | CR-VAE \lambda = 1     | 124.3     | **47** | 83.9     |
>
> **Experiment showing choice of augmentations**
> - Thank you for the suggestion. We have included the experiments of the effect of each of the augmentation individually on the MNIST dataset. It is clear to see that rotations and translations are more important than scaling, but the combination of all 3 works the best for CR-VAEs.
> |                   | MI        | AU     | NLL      |
> |-------------------|-----------|--------|----------|
> | Rotations only    | 125.8     | 45     | 82.1     |
> | Translations only | 126.1     | 45     | 81.9     |
> | Scaling only      | 125.1     | 42     | 82.7     |
> | All (CR-VAE)      | **126.3** | **47** | **81.2** |
>
> **The term of “consistency regularization ” and how that can be mistaken to think that the paper relates to frequentist statistics**
> - We note that we simply borrow the term from semi-supervised literature to ensure that we do not “reinvent the wheel” and miscredit the advances the semi-supervised learning field has made. We definitely agree that the paper is in no way related to frequentist statistics. We will clarify this in the paper.
>
> **Why did you choose the KL divergence in this direction?**
> - Technically, any divergence can be used. The idea is just to make sure the variational distribution when conditioning on the data and when conditioning on the corresponding augmentations are close. We used this form of KL (forward KL) for tractability, simplicity, and efficiency.
>
> **Another way of forcing “consistency” of the representations would be to augment the input but reconstruct the unaugmented examples (as in [4]), Have you considered this instead of your regulariser?**
> - Training the model to recover the original unaugmented sample is definitely an interesting direction and combining that idea with what we propose in this paper can be an interesting future direction. However, the idea proposed is similar to a denoising autoencoder, which does not directly solve the issue of inconsistency of representations. One consideration to be made is that it is important for a VAE to be able to reconstruct an augmented sample since if the model is deployed, it is possible that a sample resembles an augmented version of something that is in the data distribution, thereby rendering it important for the model to be able to generalize to the new data; this is the exact issue that CR-VAEs target.
>
>
>
> [1] https://arxiv.org/abs/2010.05531
> [2] https://arxiv.org/abs/1812.05941
> [3] https://arxiv.org/abs/1511.06349

---

> > ### Comment · Reviewer_svud · 2021-08-18
> > **Please add a proper discussion about learning invariance to augmentations (SSL) in your paper.**
> >
> > Thank you for all your answers.
> >
> > I think you missed one of my points on self-supervised learning. I was basically saying you are bringing to VAE's exactly what others have brought to SSL: a way of making the representations invariant to augmentations. As a result, I believe that the current method should also be compared to those SSL methods. And I have not mentioned SimCLR (which does not explicitly enforce invariance) but  [MV-IB](https://arxiv.org/abs/2002.07017) and [Barlow-twins](https://arxiv.org/abs/2002.07017) which are much closer to what you do. Are you planning on discussing these links between your method and SSL?
> >
> > I generally stand by my initial review: this is a simple and useful paper for the VAE community, but it  (1) should discuss and point out the relationship with the general idea in SSL to learn invariant representations; (2) does not really show the usefulness of the learned representations compared to strong baselines (i.e. non VAE). For the latter, I understand that the goal is to specifically improve VAE's representations, which is why I still believe that this paper is marginally above the acceptance threshold.

---

> > > ### Author Response · Authors · 2021-08-22
> > > **Thank you for your time and for checking the rebuttal**
> > >
> > > Thank you for your time to check the rebuttal!
> > >
> > > We agree with the reviewer and we will add a discussion on self supervised representation learning in the final version. We will also cite and discuss the links between invariant representation learning literature and CR-VAEs.
> > >
> > > However the method we propose is specifically to improve the representations learned by VAEs. Our goal was not to propose a method that beats all existing methods for representation learning. We are focused on VAEs. We do believe that improving VAEs fundamentally has significant value for the field since VAEs are very relevant in generating modeling, continual learning, active learning, among others. Improving VAEs can lead to significant gains in other fields as well.
> > >
> > > We will add a discussion as suggested by the reviewer in the final version.
> > >
> > > We addressed all your other comments, providing new results on ablation for different values of \lambda and the impact of the choice of augmentations. Would you improve your score in regards of this?
> > >
> > > Thanks again for your feedback.

---

> > > > ### Comment · Reviewer_svud · 2021-08-23
> > > > **Thank you but I will not improve my score.**
> > > >
> > > > Although I appreciate all the changes and think that it did improve the paper, I still do not believe that the paper should get a better score than the one I gave.
> > > > For me it is missing 3 important points, each of which would increase the score by a point (and all together would add 1 more to reach the 10):
> > > >
> > > > - **lack of novelty**: as I said, this paper is taking the regularizer from MV-IB and bringing it to VAE community. The novelty is very low. Talking about MV-IB was a prerequisite for me to keep my score, not for me to improve the score.
> > > > -**low impact outside of VAE’s**: as the authors just said, the goal of the paper is to improve VAE’s rather than providing a very good representation learning method. The latter would really improve the paper but would need comparison to SSL (and current classification results are extremely weak compared to SSL).
> > > > To be clear, focusing on marginally improving VAE’s is totally fine for a NeurIPS paper, but that definitely decreases the impact of the current paper.
> > > > - **lack of theoretical justification**: the paper is currently only (very well) written in an intuitive level. Adding theoretical justifications of how much and why exactly this can improve generative modelling would really improve the paper. Paper [4] above (“lossy compression for lossless prediction”) probably provides a good theoretical framework of invariances to understand your method.
> > > >
> > > > I still believe that it’s a simple and well written paper that should be accepted. I hope that the authors now understand what should be done for me to increase my score.

---

> > > > > ### Author Response · Authors · 2021-08-23
> > > > > **Thank you for your reply!**
> > > > >
> > > > > Thank you for your reply and for explaining your view of the paper.
> > > > >
> > > > > We understand your points and will make a pointed effort to acknowledge them and see how they can be improved upon in the final version of the paper. We will also cite [4] as suggested in the final version.

---

### Official Review · Reviewer_jSSg · 2021-07-25

**Rating:** 7
**Confidence:** 4

**Summary:**

The paper  investigates a problematic behavior of current VAE models that lies in the inconsistency of current amortized encoder networks in the sense that the latent representations of semantically similar transformations (translation, rotation etc) of image data are dissimilar. Moreover, it proposes a tractable variant of the ELBO objective that is based on data augmentation and adds a KL-regularization term that penalizes the discrepancy in the posterior of the original and the transformed image data. The VAEs trained with this objective exhibit significant benefits (in terms of the utilization of the latent space and  marginal likelihood ) when compared to naive data augmentation demonstrating sota results in image generation and classification tasks.

**Limitations And Societal Impact:**

yes

**Main Review:**

The paper is very well written and easy to follow. The new ELBO variant is intuitively justified and applicable to all exisiting VAEs. I really enjoyed reading this paper. I only have a few questions on the experiments that I feel they need to be clarified and some suggestions for further applicability of the consistency-driven ELBO variant.

(1) Could the authors make the computational demands, especially on NVAE more clear? How many GPUSs are being used? It is stated that training takes 1 day approximately for all the experiments. However, this is inconsistent with the training time reported in the NVAE paper: (Table 6 in the appendix https://arxiv.org/abs/2007.03898) the training takes 55h so, given the augmentation I would consider that the consistent vae requires ~ 100 h?? Also, how many GPU-s are you using?

(2) I also believe that especially for the deep VAEs which make use of very large latent spaces, including AU as you do in Table 1 as well as the KL divergence (how much higher is it compared to NVAE without regularization) would help clarify that the benefits come from the better use of the latent space.

(3) How many samples (S in Algorithm 1) are you using when approximating the expected value?

(4) I believe it would be really interesting (albeit not required for acceptance) if the authors demonstrate consistency on more sophisticated encoders, beyond Gaussians, such as those based on normalizing flows.

(5)In evaluation metrics, it is mentioned "we also evaluate the effect of the proposed generalization to unseen data". I think it would be very informative if the authors also demonstrate another aspect of inconsistency in VAEs-- how well it would perform on data generated by its own generative model.

**Time Spent Reviewing:**

4

---

> ### Author Response · Authors · 2021-08-10
> **Thank you for your review!**
>
> Thank you for the review! We will respond to each of your points directly in the comments below.
>
> **Computational demands for the NVAE experiments -- is it much larger than NVAE in terms of time and how many GPUs are used to train the models?**
> - Thank you for the correction. The CR-NVAE+CIFAR-10 models were trained on 8 16-GB P100 GPUs which are an older model compared to the V100 models the baseline NVAEs were trained on. The model was trained for ~4 day till completion, however it's difficult to compare the time difference between the baseline NVAE model and CR-NVAE due to the different GPU types. We will add this discussion in the final version.
>
> **Active Units of the latent space as in Table 1 for NVAE**
> - Since the NVAE paper does not provide the results for the active units for the trained NVAE models, we use the pretrained models available in the official codebase [1]. The active units were measured for the baseline NVAE for CIFAR-10 to be 211, while the CR-NVAE model was measured to be 226, suggesting improvements in the learned latent space..
>
> **Samples used in Alg 1 to evaluate the expectation**
> - For all experiments we used a batch-size of 64 and used one augmentation for each image during training. For the CR-NVAE experiments, we followed the same scheme as the original NVAE paper which uses a batch size of 200 for MNIST and 32 for CIFAR-10 (see table 6 in [2] for more specific details).
>
> **Experiments with flow architectures**
> - Using data augmentations for flow-based and other generative models, such as diffusion models, is a very promising future direction which can yield significant empirical gains. However, this paper is focused on VAEs and a simple yet effective way to improve their performance.
>
>
>
> [1] https://github.com/NVlabs/NVAE
> [2] https://arxiv.org/abs/2007.03898

---

### Decision · Program_Chairs · 2021-09-27

**Decision:**

Accept (Poster)

**Comment:**

This paper proposes an approach to improving the representations of generalization of VAEs using semantic-preserving transformations and a regularizer that enforces a congruence between representations of the original and transformed inputs.  This is shown to provide notable performance improvements for a variety of VAE models, with these gains, critically, larger than those achieved using simple data augmentation.

Overall, the reviewers were quite positive about the work, praising its simplicity, clarity, and empirical results.  Some issues were raised about the fact that most aspects of the paper are relatively straightforward applications of previous techniques, with the novelty predominantly originating only from their use in a VAE context, and insufficient discussion and comparison to other representation learning approaches that utilize semantic-preserving transformations, such as contrastive learning approaches.  However, all reviewers felt that the paper still deserved to be accepted in spite of these (albeit in some cases only marginally).

I believe that this assessment by the reviewers is quite fair and that, though not the most spectacular of papers, it is good solid work that will be of interest to the community and likely see some practical usage.  I thus recommend accepting it to the conference.

That said, I do think it is important that the authors make sure to make updates and improvements to the work, as per the reviewer's suggestions, for the camera-ready version.  In particular, I think the suggested additional discussion of related work is very important to add.  I would also strongly encourage the authors to include some non-VAE self-supervised baselines: though I mostly buy the argument that VAEs are important in their own right and do not think beating these baselines should be a condition for the work to be acceptable, I think it is still important to quantify how the approach compares to natural alternatives practitioners might consider (e.g. to see if there are aspects that this VAE-based approach is relatively good and bad at and how far off they are to SOTA approaches in the area).  Specifically, given much of the utility of the work is empirical, I feel that it is important that it properly investigates how relatively well VAE based approaches can perform in the target setting where semantically equivalent inputs can be generated.